# Protocol for the Tallaght University Hospital Institute for Memory and Cognition-Biobank for Research in Ageing and Neurodegeneration

Adam H Dyer [1,2] Helena Dolphin [1,2] Antoinette O'Connor,[3] Laura Morrison,[1,2] Gavin Sedgwick,[1] Aoife McFeely,[1,2] Emily Killeen,[1] Conal Gallagher,[1] Naomi Davey,[1] Eimear Connolly,[1] Shane Lyons,[3] Conor Young,[1] Christine Gaffney,[3] Ruth Ennis,[1] Cathy McHale,[1] Jasmine Joseph,[1] Graham Knight,[1] Emmet Kelly,[1] Cliona O'Farrelly,[4] Nollaig M Bourke,[2] Aoife Fallon,[1,2] Sean O'Dowd,[3,5] Sean P Kennelly[1,2]

AHD and HD contributed equally.

**Correspondence to**
Dr Adam H Dyer; dyera@tcd.ie

## ABSTRACT

**Introduction** Alzheimer's disease and other dementias affect >50 million individuals globally and are characterised by broad clinical and biological heterogeneity. Cohort and biobank studies have played a critical role in advancing the understanding of disease pathophysiology and in identifying novel diagnostic and treatment approaches. However, further discovery and validation cohorts are required to clarify the real-world utility of new biomarkers, facilitate research into the development of novel therapies and advance our understanding of the clinical heterogeneity and pathobiology of neurodegenerative diseases.

**Methods and analysis** The Tallaght University Hospital Institute for Memory and Cognition Biobank for Research in Ageing and Neurodegeneration (TIMC-BRAiN) will recruit 1000 individuals over 5 years. Participants, who are undergoing diagnostic workup in the TIMC Memory Assessment and Support Service (TIMC-MASS), will opt to donate clinical data and biological samples to a biobank. All participants will complete a detailed clinical, neuropsychological and dementia severity assessment (including Addenbrooke's Cognitive Assessment, Repeatable Battery for Assessment of Neuropsychological Status, Clinical Dementia Rating Scale). Participants undergoing venepuncture/lumbar puncture as part of the clinical workup will be offered the opportunity to donate additional blood (serum/plasma/whole blood) and cerebrospinal fluid samples for longitudinal storage in the TIMC-BRAiN biobank. Participants are followed at 18-month intervals for repeat clinical and cognitive assessments. Anonymised clinical data and biological samples will be stored securely in a central repository and used to facilitate future studies concerned with advancing the diagnosis and treatment of neurodegenerative diseases.

**Ethics and dissemination** Ethical approval has been granted by the St. James's Hospital/Tallaght University Hospital Joint Research Ethics Committee (Project ID: 2159), which operates in compliance with the European Communities (Clinical Trials on Medicinal Products for

## STRENGTHS AND LIMITATIONS OF THIS STUDY

⇒ Tallaght University Hospital Institute for Memory and Cognition Biobank for Research in Ageing and Neurodegeneration (TIMC-BRAiN) will prospectively recruit 1000 individuals undergoing assessment for cognitive symptoms, obtaining comprehensive clinical data and biological samples that will be stored in a secure biobank, offering a bioresource for future studies into the diagnosis, pathogenesis and management of individuals with cognitive or memory disorders.

⇒ A key strength of TIMC-BRAiN is the comprehensive neuropsychological data recorded and the extensive data capture at the time of routine clinical assessment paired with biological sampling (including blood and cerebrospinal fluid), all conducted in a routine memory assessment and support service, offering a repository of real-world patient samples and data for future studies.

⇒ A further strength is a longitudinal follow-up conducted as routine every 18 months for clinical progression, including progression to established cognitive impairment/dementia in those with early cognitive symptoms and disease severity progression in those with established dementia.

⇒ A key limitation of TIMC-BRAiN is that it is a single-site study; however, the Memory Assessment and Support Service (MASS) at Tallaght University Hospital accepts national referrals and is the largest MASS in Ireland.

Human Use) Regulations 2004 and ICH Good Clinical Practice Guidelines. Findings using TIMC-BRAiN will be published in a timely and open-access fashion.

## INTRODUCTION

The global prevalence of dementia is expected to sharply increase over the coming decades, affecting >150 million individuals globally by

2050.[1 2] Recent decades have seen significant advances in our understanding of the neurobiology of dementia, in particular, dementia due to Alzheimer's disease (AD). However, there remains an urgent need to advance understanding of the pathobiology of all dementia syndromes. Such progress will be critical to the discovery of novel diagnostic and prognostic markers, as well as advancing understanding of their real-world utility. In addition, further understanding of clinical and biological phenotypes and underlying pathophysiological processes is an urgent priority for the field to enable better approaches to personalised prevention and treatment, particularly once disease-modifying treatments become available.

Neurodegenerative diseases have high degrees of phenotypical, genetical and pathophysiological heterogeneity, with traditional diagnostic paradigms centred on late-stage syndromic classification of disease phenotypes.[3] Importantly, personalised approaches to clinical and biological classification incorporating comprehensive assessment of clinical phenotypes accompanied by the use of appropriate neuroimaging, biological sampling (such as cerebrospinal fluid (CSF) biomarkers) and further diagnostic tests serve to map observed clinical phenotypes onto known neurobiological substrates.[4 5] An accurate (and timely) diagnosis has important implications for treatment and advance care planning and is valued by those living with neurodegenerative conditions.[6 7]

Many Memory Assessment and Support Services (MASS) use diagnostic biomarkers to assist in the diagnosis and prognostication of individuals with cognitive impairment. In AD, CSF markers of amyloid and tau pathology are now frequently employed in many MASS.[8] Lumbar puncture (LP) is typically well tolerated in these settings.[9] Additionally, the recent availability of amyloid and tau positron emission tomography (PET) in certain jurisdictions allows non-invasive assessment of AD pathology.[10 11] Progress in the field of biomarker development has not been solely confined to the field of AD. Promising biomarkers that may aid in the diagnosis of other neurodegenerative disorders are also being developed, for instance, the identification of isoform-specific tau species in primary tauopathies, such as corticobasal degeneration and progressive supranuclear palsy (PSP), and the detection of alpha-synuclein in CSF by real-time quaking-induced conversion in prodromal Parkinson's disease and dementia with Lewy bodies.[12 13]

One of the most exciting advances in the field of neurodegenerative disorders has been the advent of blood-based biomarkers. At present, blood-based biomarkers—exemplified by research in AD—show remarkable promise, even in presymptomatic disease stages.[14–19] These advances will be accompanied by significant challenges in the implementation and interpretation of these biomarkers in clinical practice. Importantly, validation of these biomarkers in clinical populations, with consideration of issues such as renal clearance, medical comorbidity and population-specific norms, will require the use of large validation cohorts prior to widespread implementation.

A key priority in dementia research is understanding clinical and biological phenotypes across the disease spectrum, which can aid in accurate diagnosis, prognostication and treatment plans for those affected. For instance, mild cognitive impairment (MCI) is characterised by deficits evident on neuropsychological testing that do not interfere with day-to-day function and may stabilise, progress to dementia or even revert to normal cognition over time.[20] Around 5%–15% of individuals with MCI progress to dementia annually.[21] One of the greatest challenges at present is predicting the variable disease trajectory in those affected by MCI, and there is an urgent need for new clinical, diagnostic and biological markers that may indicate a greater likelihood of disease progression. While several notable studies have demonstrated the influence of AD biomarkers on disease progression in MCI, further studies are needed to establish the optimal use of prognostic markers in individuals with early cognitive impairment.[22]

The development of large-scale clinical, imaging, genetical and biological repositories and real-world clinical cohorts is crucial to the discovery, identification, validation and standardisation of clinical and biomarker-based assessments for AD and other neurodegenerative conditions.[15 23–26] Longitudinal observational cohort studies are integral to advancing understanding of the relationship between clinical and fluid biomarkers, cognition and clinical progression across all neurodegenerative diseases, especially as we enter an era of disease-modifying therapies.[27] The development of further large cohorts is essential in facilitating the translation of biomarker-based discoveries into clinical practice and further research into the underlying pathobiology and treatment of neurodegenerative conditions.

The Tallaght University Hospital Institute of Memory and Cognition Biobank for Research in Ageing and Neurodegeneration (TIMC-BRAiN) will create a longitudinal biobank of clinical data and biological samples in individuals undergoing assessment for memory and cognitive symptoms at TIMC-MASS. This biobank will facilitate research into clinical and biological biomarkers, in addition to research studies focused on the underlying neurobiology of MCI, dementia and other neurodegenerative conditions.

## AIMS

TIMC-BRAiN is a longitudinal biobanking study commencing in January 2023 that will recruit 1000 individuals attending the TIMC-MASS for workup and assessment of cognitive symptoms. TIMC-BRAiN will biobank longitudinal clinical and neuroimaging data alongside biological samples. This biobank will serve as a repository for future research studies that seek to (1) advance understanding of disease pathobiology, (2) evaluate the use of diagnostic and prognostic tests/assessments and (3) improve precision medicine approaches across neurodegenerative diseases.

The aims of TIMC-BRAIN are as follows:

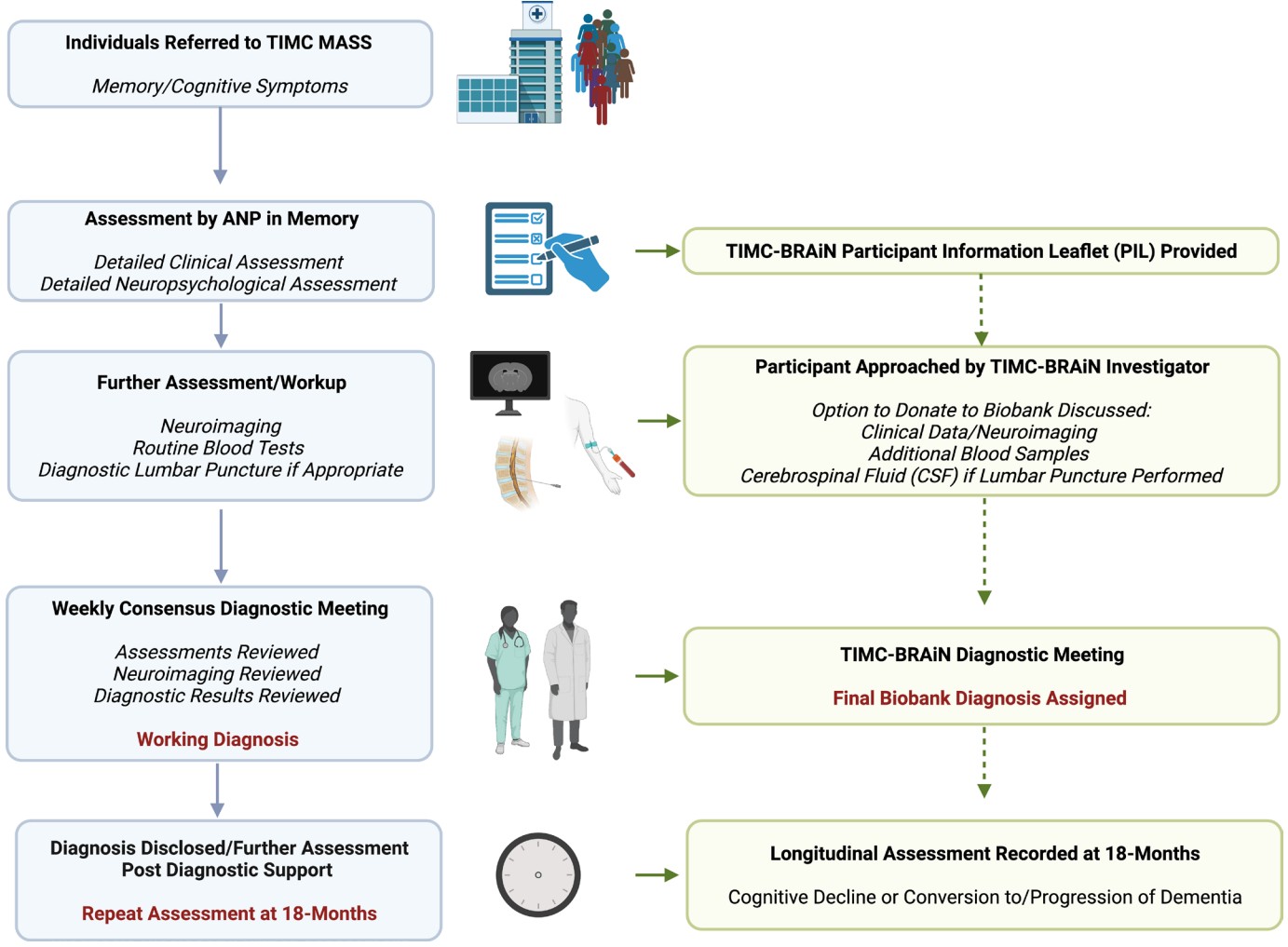

**Figure 1** Recruitment workflow for TIMC-BRAiN. Individuals are referred to the TIMC-MASS for concerns over cognition or memory symptoms and are first assessed by an ANP in memory. For TIMC-BRAiN, participants undergoing assessment will be provided with a PIL to consider donating clinical and/or biological samples to the TIMC-BRAiN biobank study. Should individuals wish to participate, they will undergo informed consent and may donate biological samples at the time of a diagnostic lumbar puncture or alongside routine phlebotomy. Participants are followed up at 18 months to examine conversion and clinical progression. ANP, advanced nurse practitioner; PIL, patient information leaflet; TIMC-BRAiN, Tallaght Institute of Memory and Cognition Biobank for Research in Ageing and Neurodegeneration; TIMC-MASS, Tallaght Institute of Memory and Cognition Memory Assessment and Support Service.

▶ To create clinical data, neuroimaging and biological sample repositories from individuals being assessed for concerns relating to cognition at the TIMC-MASS.

▶ To record a final diagnosis and comprehensive clinical phenotyping in those recruited, including demographic information, medical history, cardiovascular risk factors, family history, specific cognitive symptoms (for instance, episodic memory/autobiographical memory, language, facial recognition and topographical memory), neuropsychological symptoms, mobility/gait, sleep, nutrition, mood, frailty, hearing and vision.

▶ To record neuroimaging results coupled to clinical data consisting of MRI or CT brain results with scoring of MRI scans for vascular burden, parietal atrophy and medial temporal lobe atrophy in addition to results

of nuclear imaging and other scans performed where clinically appropriate.

▶ To biobank biological samples including peripheral blood (whole blood for DNA, serum, plasma and whole blood stored in blood stabiliser) and CSF, including both CSF supernatant and cryopreserved immune cells.

▶ Longitudinally track changes in cognition (via repeat cognitive assessment) and conversion to/progression of dementia at subsequent 18-month follow-up visits, performed alongside routine clinical care.

As TIMC-BRAiN is a longitudinal clinical data and biological sample repository, there are no prespecified hypotheses to be tested. TIMC-BRAiN will afford a ready clinical and biological repository for important research questions to be answered in the future and aims to

collaborate widely to enable novel basic biological and translational research aimed at improving diagnosis, prognostication and treatments for those affected by neurodegenerative diseases.

## METHODS
### Study setting and design
Tallaght University Hospital (TUH) is a tertiary referral hospital in Dublin, Ireland, with a catchment area of nearly 500 000 individuals. The TUH Institute of Memory and Cognition (TIMC) is the largest MASS in Ireland and receives referrals from both the TUH catchment and nationally for individuals experiencing cognitive symptoms. Approximately 400–500 patients are assessed annually in the TIMC-MASS. Once referred to the service, patients are assessed in the first instance by an advanced nurse practitioner (ANP). The comprehensive assessment includes detailed neuropsychological assessment, medical assessment, routine blood tests and appropriate neuroimaging before each case is individually discussed at a weekly multidisciplinary team consensus meeting where the working diagnosis is discussed and further investigations, if needed, are advised. We anticipate that TIMC-BRAiN will recruit approximately 200 individuals per year to donate clinical data and biological samples, with an estimated 100 of these coming from the roughly 150 individuals who undergo routine diagnostic LP annually (paired CSF and blood) and 100 individuals undergoing assessment without LP (blood only) from the roughly 250 assessed annually in our unit.

### Inclusion and exclusion criteria
TIMC-BRAiN will recruit participants undergoing assessment and workup in the TIMC-MASS (see figure 1) who are willing to provide biological samples for research. TIMC-BRAiN will recruit a diverse cohort, including individuals with different levels of cognitive performance—subjective memory concerns (SMCs), MCI and dementia—and a representative array of neurodegenerative diseases—to include AD, Lewy body disease (LBD), frontotemporal dementia (FTD) and other neurodegenerative conditions. In order to reflect a real-world clinical cohort, participants will not be excluded from TIMC-BRAiN based on age or medical comorbidity. However, individuals with severe systemic illnesses such as malignancy with limited life expectancy, individuals with current significant alcohol or substance misuse or those with current significant psychiatric comorbidity will be excluded from participation.

Participants will be provided with a participant information leaflet at the time of routine clinical assessment with the ANP in the TIMC and allowed to reflect on whether they wish to donate their clinical/neuroimaging data and biological samples to the TIMC-BRAiN biobank. Participants are informed that their participation is entirely voluntary and will not, in any way, affect their future clinical care. For those undergoing LP, participants are approached by a TIMC-BRAiN investigator/subinvestigator at the time of the procedure to assess whether they wish their clinical data (obtained at the time of assessment with the ANP) or biological samples to be included in the biobank. For those not undergoing LP, a separate appointment for phlebotomy only is arranged if individuals wish to part-take and have their clinical data and biological samples (blood) biobanked. Phlebotomy is incorporated alongside routine phlebotomy where possible to minimise venepuncture. The outline assessment and recruitment pathways are shown in figure 1. The Participant Information Leaflet and Consent Form are provided in online supplemental file 1. Investigators and subinvestigators consist of medical staff (specialist registrars, clinical fellows and consultants in geriatric medicine and neurology) who are part of the TIMC and are involved in providing clinical care. It is emphasised to individual participants that their willingness to take part (or not) in TIMC-BRAiN does not affect their clinical care in any way.

### Clinical and neuropsychological assessments
Clinical and cognitive assessments are performed by an ANP as part of routine care at the TIMC. The contents of the TIMC case report form (CRF) are given in table 1. Information collected includes background/demographic information, medical history, regular medications, hearing and vision, smoking status (yes/no/previous) and family history (detailed in table 1).

All participants undergo a comprehensive clinical history, a multidomain cognitive assessment and a collateral history, which specifically examines driving safety and medication compliance in addition to classifying duration and domains of cognitive change. Cognitive symptoms are assessed by recent changes in the following domains: (1) episodic memory, (2) autobiographical memory, (3) language and (4) facial recognition/topographical memory. All of these variables are recorded as yes/no to indicate a recent change. The total duration of symptoms (in months) and the first predominant symptom are also recorded.

Neuropsychological assessment consists of Addenbrooke's Cognitive Examination[28] and Frontal Assessment Battery[29]; these items are completed by all participants. Additionally, the Repeatable Battery for the Assessment of Neuropsychological Status is performed if possible.[30] The Clinical Dementia Rating Scale global/sum of boxes is used to assess for dementia severity.[31] The Ascertain Dementia 8 questionnaire is also routinely administered to informants.[32] The Cambridge Behavioural Inventory captures recent behavioural changes, cognitive changes and affective symptoms are reported.[33]

Mood and anxiety are routinely assessed via the Hospital Anxiety and Depression Scale.[34] Sleep is assessed via the Pittsburgh Sleep Quality Index.[35] Mobility is assessed by the Timed-Up-and-Go test.[36] Frailty is assessed via the Clinical Frailty Scale.[37] Nutrition is assessed in all participants using the Mini-Nutritional Assessment.[38] The results of all these assessments are recorded in the TIMC-BRAiN CRF.

**Table 1** Information recorded on TIMC-BRAiN case report form

| Item | Data recorded |
|---|---|
| **Demographic information** | |
| Age | Age at assessment (years) |
| Sex | Biological sex (male/female/non-binary) |
| Level of education | Finished formal education (years) |
| Occupation | Occupation (free text) |
| Age retired | Age at retirement (years) |
| Marital status | Non-married/married/divorced/widowed |
| Driving status | Currently driving/ceased driving/never drove |
| **Medical history** | |
| Prior stroke/transient ischaemic attack | Yes/no |
| Recurrent syncope | Yes/no |
| Diabetes mellitus[5] | History of/on diabetes mellitus medication; yes/no |
| Hypertension | History of/on antihypertensive medication; yes/no |
| Hypercholesterolaemia | History of/on antilipidaemic medication; yes/no |
| Ischaemic heart disease | Yes/no |
| Alcohol excess | Yes/no |
| Epilepsy | History of/on antiepileptic medication; yes/no |
| Concussion/prior head injury | Yes/no |
| History of malignancy | Yes/no |
| Previous anxiety | History of/on anxiolytic medication; yes/no |
| Previous depression | History of/on antidepressant medication; yes/no |
| Hearing impairment | Yes/no |
| Vision impairment | Yes/no |
| Smoking status | Yes/no/previous smoker |
| Family history | Memory difficulties/dementia/Alzheimer's disease/in a first-degree relative; yes/no |
| Regular medications | Medications list; coded using anatomical therapeutic classification system |
| Anosmia | Yes/no |
| **Cognitive symptoms at presentation** | |
| Symptom duration | Duration of symptoms (in months) |
| Episodic memory | Each item recorded as yes/no indicating recent change at time of assessment |
| ► Lose track of days<br>► Forgetting appointments<br>► Misplacing objects<br>► Forgetting to pay bills<br>► Forgetting how to use appliances<br>► Remembering to take medications | |
| Autobiographical memory | Recorded as yes/no indicating recent change at time of assessment |
| ► Forgetting past personal events | |
| Language | Each item recorded as yes/no indicating recent change at time of assessment |
| ► Word-finding difficulties<br>► Shrinkage of vocabulary<br>► Comprehending Speech<br>► Comprehending written information<br>► Ability to engage in conversations | |

**Table 1** Continued

| Item | Data recorded |
|------|---------------|
| Recognition | Each item recorded as yes/no indicating recent change at time of assessment |
| ▶ Facial recognition<br>▶ Getting lost in familiar areas<br>▶ Changes in using public transport/driving | |
| Informant concerns | Each item recorded as yes/no indicating recent change at time of assessment |
| ▶ Issues with cooking<br>▶ Issues with orientation<br>▶ Issues with driving<br>▶ Issues with medication compliance<br>▶ Recent falls | |
| **Cognitive assessment** | |
| RBANS | Repeatable Battery for Assessment of Neuropsychological Status<br>Unadjusted (Raw) score and centile score computed (based on age/education)<br>Normalisation using Duff norms |
| ▶ Index I (immediate memory)<br>▶ Index II (visuospatial/constructional)<br>▶ Index III (language)<br>▶ Index IV (attention)<br>▶ Index V (delayed memory)<br>▶ Global score | |
| ACE-III | Addenbrooke's Cognitive Assessment if unable to complete RBANS<br>Domain and total scores recorded |
| ▶ Attention<br>▶ Memory<br>▶ Fluency<br>▶ Language<br>▶ Visuospatial<br>▶ Overall score | |
| CDR | Clinical Dementia Rating Scale<br>Global score and sum of boxes recorded |
| ▶ Memory<br>▶ Orientation<br>▶ Judgement and problem solving<br>▶ Community affairs<br>▶ Home and hobbies<br>▶ Personal care | |
| FAB | Frontal Assessment Battery<br>Total score recorded |
| AD8 administered to informant | Ascertain-Dementia 8 Questionnaire (scored from 8) |
| CBI-R | Cambridge Behavioural Inventory-Revised |
| **Mood and anxiety assessment** | |
| HADS-Depression<br>HADS-Anxiety | Hospital Anxiety and Depression Scales<br>Total Score Recorded<br>Probable Depression/Anxiety (Score >10) Recorded |
| **Further clinical assessment** | |
| PSQI | Pittsburgh Sleep Quality Index (0–21) |
| TUG | Timed-Up-and-Go Test; Time to Complete in seconds recorded |
| CFS | Clinical Frailty Scale to Assess Frailty |
| MNA | Mini Nutritional Assessment Score (0–7: malnourished; 8–11: at risk and 12–14: normal) |
| **Neuroimaging** | |
| MRI brain | Adjudicated at MDT consensus meeting by a panel of more than two geriatricians/neurologists<br>▶ Fazekas Score 0–3 for white matter disease<br>▶ MTA Score 0–4<br>▶ Koedam Score for Parietal Atrophy Score 0–3 |
| ▶ Fazekas Score<br>▶ MTA Score<br>▶ Koedam Score | |

Continued

**Table 1** Continued

| Item | Data recorded |
|---|---|
| CT brain results | CT brain results if performed; free text |
| FDG-PET results | FDG-PET results if performed; free text |
| DAT results | DAT scan results if performed; free text |
| Blood test results | |
| Haematology results | Any abnormalities detected; yes/no; recorded in standard units as per hospital laboratory |
| ▶ Full blood count | |
| Biochemistry and other results | Any abnormalities detected; yes/no; recorded in standard units as per hospital laboratory |
| ▶ Renal profile<br>▶ Liver profile<br>▶ Bone profile<br>▶ Glycated haemoglobin<br>▶ Lipid profile<br>▶ Micronutrients: vitamin B12 and folate<br>▶ Vitamin D | |
| Cerebrospinal fluid results | |
| Diagnostic lumbar puncture results | All recorded as pg/mL using Roche Elecsys electrochemiluminesence assay |
| ▶ $A\beta_{1-42}$<br>▶ T-Tau<br>▶ P-Tau | |
| Working (consensus meeting) diagnosis | |
| Working diagnosis at weekly MDT meeting | Free text |
| Final (biobank) diagnosis | |
| Final diagnosis adjudicated following all investigations | Functional status |
| | ▶ Subjective memory complaints<br>▶ Mild cognitive impairment<br>▶ Dementia |
| | Aetiological diagnosis |
| | ▶ Alzheimer's disease (amnestic, behavioural, LPA, CBS and PCA variants)<br>▶ Lewy body disease<br>▶ FTD (bvFTD/nfvPPA/svPPA)<br>▶ FTD overlap syndromes: FTD-ALS/FTD-CBS/FTD-PSP<br>▶ CBS: 4R tauopathy<br>▶ PSP<br>▶ Vascular cognitive impairment/dementia<br>▶ Other diagnosis (free text)<br>▶ Genetic diagnosis (free text) |

ACE, Addenbrooke's Cognitive Assessment; ALS, amyotrophic lateral sclerosis; bvFTD, behavioural variant FTD; CBS, corticobasal syndrome; DAT, dopamine transporter scan; FDG, fluorodeoxyglucose; FTD, frontotemporal dementia; LPA, logopenic variant aphasia; MDT, multidisciplinary team; MTA, medial temporal atrophy; nfvPPA, non-fluent agrammatic PPA; PCA, posterior cortical atrophy; PET, positron emission tomography; PPA, primary progressive aphasia; PSP, progressive supranuclear palsy; 4R Tau, 4 repeat tauopathy; svPPA, semantic variant PPA.

## Neuroimaging

All individuals assessed in the TIMC have an MRI brain performed unless contraindicated. MRI scans are reviewed at weekly consensus meetings and the following scores are applied:

i. Fazekas Score: grades white matter disease, scored from 0 to 3.[39]
ii. Medial temporal lobe atrophy: grades mesiotemporal atrophy, scored from 0 to 4.[40]
iii. Parietal Atrophy Score (Koedam Score): grades parietal atrophy, scored from 0 to 3.[41]

Scores are applied by a panel of at least two consultants in geriatric medicine and neurology. These scores are recorded in the TIMC-BRAiN data repository. Some individuals proceed to have additional imaging, for example, fluorodeoxyglucose-PET or Dopamine uptake scans. The outcome of all relevant imaging is documented within the data repository (see table 1).

## Diagnostic workflow in TIMC-MASS and final TIMC-BRAIN diagnosis

All cases, after completion of their clinical, neuropsychological and neuroimaging assessments, are discussed at a weekly consensus diagnostic meeting. This meeting, led by a consultant geriatrician/neurologist with expertise in neurodegenerative diseases, determines, where possible, a working (consensus) diagnosis. Further investigations, such as advanced neuroimaging or diagnostic CSF sampling, are recommended in appropriate cases.

For individuals recruited into TIMC-BRAiN, a separate biobank diagnostic meeting occurs on a bimonthly basis. This meeting is convened by the biobank coordinator and attended by the biobank coordinator, data manager, research fellows, consultant geriatrician, consultant neurologist and study subinvestigators. Each case is discussed only after all appropriate clinical investigations/assessments have taken place and a final 'biobank' diagnosis is confirmed reflecting (1) functional status and (2) aetiological diagnosis as follows.

### Functional status

► SMCs: individuals with concerns over cognition/memory symptoms, free from established cognitive impairment on neuropsychological testing.

► MCI: impairments on one or more neuropsychological domains (typically 1.5 SD from age/education mean) that do not interfere with day-to-day function.[20]

► Dementia: objective cognitive loss of any domain that is severe enough to cause a functional decline in day-to-day activities.

### Aetiological diagnosis

Published consensus criteria are used to classify aetiology as:

► AD (subtyped into amnestic, behavioural, logopenic variant aphasia, corticobasal syndrome (CBS) and posterior cortical atrophy variants).[5 42 43]

► LBD.[44]

► FTD subtyped into behavioural variant FTD, semantic variant FTD and non-fluent agrammatic FTD[45 46] Also, where applicable, include overlap syndromes: FTD-CBS, FTD-PSP and FTD-amyotrophic lateral sclerosis.

► CBS: 4R tauopathy.[47 48]

► PSP.[49]

► Vascular cognitive impairment/dementia,[50]

► Other diagnosis: this includes individuals with a diagnosis that does not conform to one of the above categories. Details around potential diagnoses will be recorded as free text by the biobank manager.

► Genetic diagnosis: for the subset of participants with a genetic diagnosis, this will also be reordered.

In cases where participants meet the criteria for two aetiological diagnoses, for example, non-fluent agrammatic primary progressive aphasia and FTD-PSP, both will be recorded as the final TIMC-BRAiN diagnosis. The biobank coordinator is responsible for recording the final biobank diagnosis in the appropriate participant's Research Electronic Data Capture (RedCAP) diagnostic section (see the section 'Data Storage').

### Blood and CSF sampling

Following informed consent, blood sampling will be performed as part of routine phlebotomy if possible but in certain cases, it may need to be performed separately. All blood tests will be obtained between 08:00 and 12:00 to minimise the risk of diurnal variation on biomarkers and processed on-site at TIMC within 2 hours of the blood draw. For TIMC-BRAiN participants, a serum clot activator tube (9 mL), two EDTA tubes (9 mL each) and a lithium heparin-coated tube (3 mL) are collected. Fasting status will be recorded; however, participants are not asked to fast for the purposes of TIMC-BRAiN participation.

Individuals undergoing diagnostic CSF examination as part of their clinical assessment will be offered the opportunity to donate CSF samples to the TIMC-BRAiN biobank by a study investigator/subinvestigator. LPs are performed between 09:00 and 12:00 and are carried out in a standard aseptic fashion. LPs are performed at the L3–L5 level and manometers are avoided. The maximum number of attempts (including changes of operator where required) is 3. Diagnostic samples are obtained in the first instance, followed by an additional 5–10 mL of CSF in 2.5 mL sterile polypropylene tubes (Starstedt, Cat no. 63.614.625) for donation to the TIMC-BRAiN biobank. CSF is collected by the drip method (gravity drip) as standard. Samples are inverted three times and processed on-site within 30 min of collection.

### Sample processing for TIMC-BRAiN

See figure 2 for blood and CSF sample processing protocols. Blood and CSF samples are processed on-site at TIMC as soon as possible after collection. Whole blood (0.5 mL) is pipetted from one of the EDTA tubes and stored in a sterile polypropylene cryovial for later DNA analysis. One millilitre of blood is removed from the lithium heparin tube and stored in 2×0.5 mL aliquots with 0.5 mL of Cytodelics whole blood stabiliser for potential immune cell analysis by flow cytometry.[51] Following this, the remaining blood samples (2x EDTA tubes and 1x serum clot activator tubes) are centrifuged at 1.8xg for 10 min. Plasma and serum are subsequently aliquoted in 0.5 mL sterile cryovials, labelled with an anonymous TIMC-BRAiN participant ID and stored alongside whole blood and Cytodelics aliquots at −80°C for future analysis.

CSF samples are centrifuged at 400×g for 10 min at 4°C to pellet immune cells. Cell-free CSF is subsequently aliquoted into 0.5 mL sterile cryovials and stored at −80°C for future analysis. The remaining pellet is resuspended in 0.9 mL Recovery Cell Culture Freezing Medium (ThermoScientific) and stored overnight in a Mr Frosty Freezing Container at −80°C with aliquots removed and stored separately the following day for future use. This protocol is based on previously published reports examining immune cell composition.[52 53]

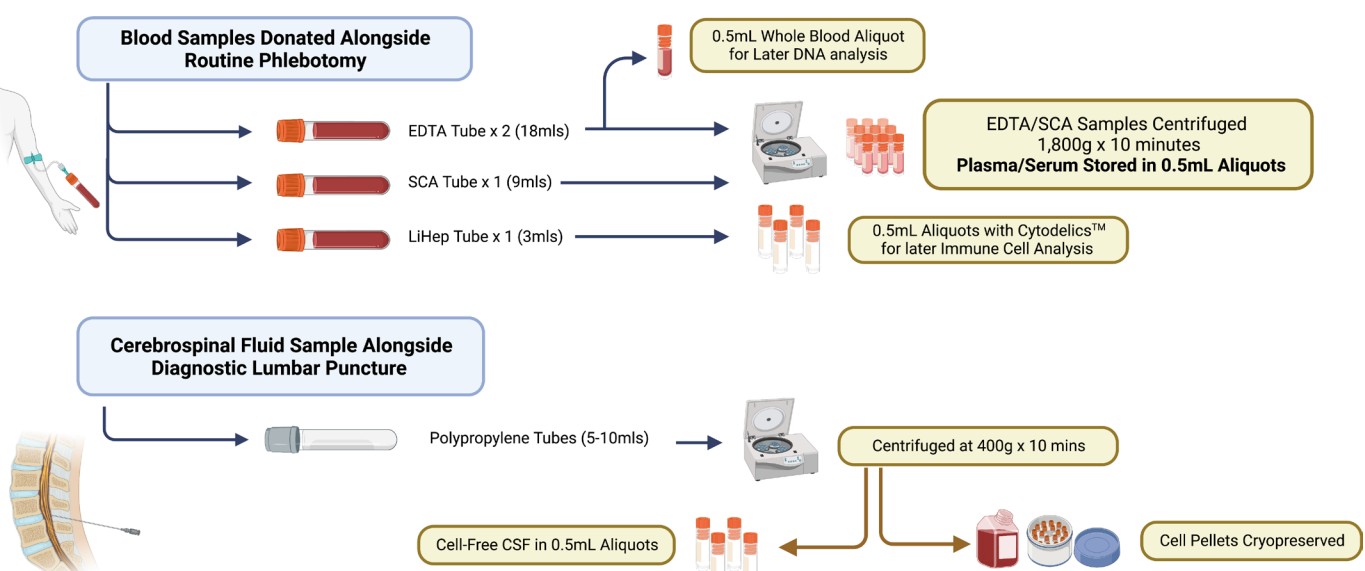

**Figure 2** Biological sample processing for the Tallaght Institute of Memory and Cognition Biobank for Research in Ageing and Neurodegeneration. Biological samples are obtained either alongside routine phlebotomy (for blood samples) or at the time of diagnostic lumbar puncture (for CSF where performed). CSF, cerebrospinal fluid; LiHep, lithium heparin; SCA, serum clot activator.

## Data storage

For each participant, a unique anonymous TIMC-BRAiN ID will be generated that links clinical/neuroimaging data with biological samples. The clinical and cognitive data obtained will be recorded on a dedicated TIMC-BRAiN CRF once a final biobank diagnosis has been applied. A small subset of variables is also recorded for the date of blood/CSF sampling and storage in the biobank on a study sample storage form. The CRF is completed by a study investigator/subinvestigator following a final biobank diagnosis and aims to capture all data obtained as part of routine clinical assessment, as outlined above. Data will be stored using RedCAP, a web application for building and managing online surveys and databases for research studies. An institutional RedCAP system, protected by host and institutional firewalls, will be maintained at TUH by the TIMC-BRAiN study team. Only the principal investigator, study investigators/subinvestigators and biobank manager will have complete access to the TIMC-BRAiN database on RedCAP. Biological samples that are coded to match the SSF on RedCAP are stored in a dedicated freezer space in the Meath Foundation Laboratory, TUH. As part of the TIMC-BRAiN consent procedure, participants give permission for the storage of data for an initial period of 5 years, which may be extended indefinitely according to review by the TIMC-BRAiN steering committee.

## Longitudinal data

As part of routine care at TIMC, participants diagnosed with MCI or dementia are routinely followed up in postdiagnostic support pathways. As part of this, all participants are regularly reviewed with a routine repeat cognitive/neuropsychological assessment and dementia severity rating at 18 months. This data will be available for TIMC-BRAiN participants electing to have their clinical data

stored in the TIMC-BRAiN repository. A follow-up CRF will document changes in cognitive/neuropsychological test scores in addition to progression to dementia/progression of dementia severity.

## Withdrawal procedure

If a participant wishes to withdraw from the study at any stage, this decision is clearly documented by the TIMC-BRAiN investigator/subinvestigator in the patient's medical notes and CRF. Participants' clinical data will be fully removed from the RedCAP database, including CRF and SSF. Biological samples donated will be identified by the biobank manager and destroyed. Participants will be informed that the decision to withdraw from TIMC-BRAiN will not affect their ongoing care.

## Oversight

The TIMC-BRAiN biobank is guided by a steering committee with a full meeting convened four times per year and reviews any requests for collaboration/sample use in addition to reviewing the day-to-day operating procedures of the biobank.

## Data and sample access

Applications for access to clinical data/neuroimaging data or biological samples from the TIMC-BRAiN biobank are considered by the Biobank committee and granted for specific reasons. Applications are reviewed once per quarter at the committee meeting and are reviewed, held pending further information/discussion or rejected based on study feasibility or quality, keeping in mind the efficient use of the finite sample repository of TIMC-BRAiN and avoiding duplication of research effort. Following request approval, samples and accompanying clinical/neuroimaging data are dispensed to

the requesting study team via the Biobank Manager and recorded on the TIMC-BRAiN recruitment log. For the transfer of samples and data outside of the host institution (TUH), a Materials Transfer Agreement will be generated, reviewed at the Biobank committee meeting and signed by both institutions.

## Public and patient involvement

Despite the fact that the majority of older adults presenting to geriatric medicine services are interested in participating in research,[54] older adults are typically underrepresented in cohort studies and clinical trials, with many imposing arbitrary age-related or medical comorbidity-based cut-offs.[55] To ensure the relevance, acceptability and feasibility of the TIMC-BRAiN biobank, feedback on the protocol design was obtained from patient representatives undergoing diagnostic workups in the TIMC-MASS. Additionally, a patient representative sits on the TIMC-BRAiN steering committee, which discusses the ongoing use of research samples, questions to be addressed and the day-to-day running of the biobank.

## DISCUSSION

Over a 5-year period, the TIMC-BRAiN biobank will aim to create a large clinical data and biological sample repository to facilitate basic research aimed at understanding the underlying pathobiology of dementia, the discovery and validation of novel diagnostic and prognostic markers and facilitate research aimed at elucidating potential new treatment options for individuals living with neurodegenerative disease. By creating a central data repository of anonymised clinical data and biological samples, TIMC-BRAiN will enable suitable research questions to be addressed by accessing a ready and comprehensively phenotyped repository. The availability of such a repository will facilitate novel research questions into the underlying aetiology, diagnosis and treatment of neurodegenerative disease and speed up collaborative research efforts for those living with neurodegenerative disease.

TIMC-BRAiN will recruit 1000 individuals over 5 years and obtain comprehensive clinical data and biological samples that will be stored in a secure biobank, offering a bioresource for future studies. One of the key strengths of TIMC-BRAiN is the comprehensive neuropsychological data recorded and the extensive data capture at the time of routine clinical assessment paired with biological sampling (including blood and CSF), all conducted in a routine memory assessment and support service, offering a repository of real-world patient samples and data for future studies. This is crucial in the validation of new biomarkers and diagnostic tests as well as the potential evaluation of new hypotheses around the underlying pathobiology of memory disorders and dementia in older adults. TIMC-BRAiN is also notable for its incorporation alongside clinical care as routine, where follow-up is currently conducted as routine every

18 months for clinical progression, including progression to established cognitive impairment/dementia in those with early cognitive symptoms and disease severity progression in those with established dementia. While TIMC-BRAiN is a single-site study, the MASS at TUH accepts national referrals and is the largest MASS in Ireland. TIMC-BRAiN is the first biobank embedded within a MASS in Ireland.

With the advent of new CSF and blood biomarkers in neurodegenerative disease, it is envisaged that TIMC-BRAiN will offer an invaluable resource to research projects aimed at further validating these markers across different populations and at different stages of the disease process. Crucial to this is the real-world nature of the TIMC-BRAiN cohort, comprised of individuals undergoing diagnostic workup for early cognitive symptoms or memory complaints. This is crucial to understanding the real-world utility of novel biomarkers for neurodegenerative disease as they emerge. Additionally, by providing basic biomedical researchers with access to anonymised patient blood and CSF samples to readily use, TIMC-BRAiN aims to act as a catalyst for facilitating new insights into the pathobiology of neurodegenerative diseases.

## ETHICS AND DISSEMINATION

Ethical approval has been granted by the St. James's Hospital/TUH Joint Research Ethics Committee which operates in compliance with the European Communities (Clinical Trials on Medicinal Products for Human Use) Regulations 2004 and ICH Good Clinical Practice Guidelines (Project ID: 2159). Written informed consent will be obtained from all individuals donating clinical/biological samples to TIMC-BRAiN. A data protection impact assessment and review was performed as part of this application at TUH and TIMC-BRAiN is fully compliant with General Data Protection Regulations.

For the dissemination of research findings, this will be conducted in several ways. In the first instance, results will be presented at relevant national/international conferences in the areas of geriatric medicine, neurology, memory disorders and dementia. Further, data from studies using data or samples from TIMC-BRAiN will be presented locally and within the host institution. All collaborators who make a meaningful contribution to the recruitment, assessment, data curation and management of TIMC-BRAiN will be included in published outputs. Results from studies conducted using the TIMC-BRAiN biobank will also form part of peer-reviewed journal publications and in doing so, it is a key priority for the TIMC-BRAiN biobank that all papers arising from the biobank are published in an open-access fashion.

**Author affiliations**
[1]Institute of Memory and Cognition, Tallaght University Hospital, Dublin, Ireland
[2]Discipline of Medical Gerontology, School of Medicine, Trinity College Dublin, Dublin, Ireland
[3]Department of Neurology, Tallaght University Hospital, Dublin, Ireland

[4]Comparative Immunology, Trinity College Dublin, Dublin, Ireland
[5]Academic Unit of Neurology, Trinity College Dublin, Dublin, Ireland

**Acknowledgements** The authors wish to acknowledge support from the TIMC-MASS clinical and administrative staff.

**Contributors** AHD, HD and SPK are responsible for the overall design, administration and conduct of the TIMC-BRAiN biobank. AHD and HD jointly wrote the manuscript. AOC, LM, GS, AM, EKi, CGal, ND, EC, SL, CY, CGaf, RE, CM, JJ, GK, Eke, AF and SOD designed clinical protocols for the assessment of participants. AHD, HD, COF, NMMB and SPK designed the laboratory protocols. All authors have read and approved the final manuscript. All authors were involved in informing the aims and design of the study.

**Funding** The TIMC-BRAiN Biobank, AHD, HD and SPK are funded by the Meath Foundation, Tallaght University Hospital. AHD has been awarded the Irish Clinical Academic Training (ICAT) Programme, supported by the Wellcome Trust and the Health Research Board (Grant Number 203930/B/16/Z), the Health Service Executive, National Doctors Training and Planning and the Health and Social Care, Research and Development Division, Northern Ireland.

**Competing interests** None declared.

**Patient and public involvement** Patients and/or the public were involved in the design, or conduct, or reporting, or dissemination plans of this research. Refer to the Methods section for further details.

**Patient consent for publication** Not applicable.

**Provenance and peer review** Not commissioned; externally peer reviewed.

**ORCID iDs**
Adam H Dyer http://orcid.org/0000-0003-1356-510X
Helena Dolphin http://orcid.org/0000-0001-5540-7850

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
