## [Reviewer comments · BMJ Open]

ARTICLE DETAILS

TITLE (PROVISIONAL)	Protocol for the Tallaght University Hospital Institute for Memory and Cognition – Biobank for Research in Ageing and Neurodegeneration (TIMC-BRAiN)
AUTHORS	Dyer, Adam; Dolphin, Helena; O'Connor, Antoinette; Morrison, Laura; Sedgwick, Gavin; McFeely, Aoife; Killeen, Emily; Gallagher, Conal; Davey, Naomi; Connolly, Eimear; Lyons, Shane; Young, Conor; Gaffney, Christine; Ennis, Ruth; McHale, Cathy; Joseph, Jasmine; Knight, Graham; Kelly, Emmet; O'Farrelly, Cliona; Bourke, Nollaig M; Fallon, Aoife; O'Dowd, Sean; Kennelly, Sean P

VERSION 1 – REVIEW

REVIEWER	Emma Cunningham Queen's University Belfast, Centre for Public Health
REVIEW RETURNED	29-Sep-2023

GENERAL COMMENTS	This manuscript helpfully describes in detail the planned set up of a Biobank and will be of interest to clinicians and researchers considering similar endeavours and also for reviewers and readers of subsequent outputs from this Biobank. I have a few suggestions: 1. Information regarding the recruitment timeline could be made more clear. The text suggests people are approached at the time of LP/phlebotomy but it also says phlebotomy can be undertaken for the sole purpose of Biobanking. It would seem likely that phlebotomy occurs on the same day as the clinical assessment that all the clinical data is collected at, are people approached then? Is inclusion based on agreeing to donate a blood or CSF sample - I assume this is the case but this should be explicitly stated. Who are the study investigators/sub-investigators, are they members of the direct clinical care team?2. It would be nice if the consent form and if possible the Participant Information Leaflet could be included as supplementary material.3. The text below Table 1 includes clinical information not included in Table 1. Conversely, page 14 line 46 to page 15 line 13 describe assessments that are all included in Table 1 and the text could be omitted if the references were included in the Table.4. Would the authors consider including information regarding physical activity, marital status and social support services (this may well be evident from the other information collected but I cannot see it specifically recorded in Table 1)? Driving is also not listed in Table 1 but is mentioned in the text below.5. Will there be the option to retrospectively interrogate participants' medical records to find additional information if this is required subsequent to the baseline and/or 18 month follow up visit? For example if routine blood biochemistry results are recorded as abnormal yes/no would it be possible to retrieve GFR at the time of
--

	blood-based biomarker sampling subsequently? Are there any electronic health records that the Biobank will link to? 6. The authors note they assess 400-500 patients per year and anticipate recruiting 1000 people over 5 years ie 40-50% of those attending memory clinic. Given their wide eligibility criteria can they comment on this anticipated figure. It is a very reasonable figure but would be helpful to understand the reasons the authors think people might not be approached (eg if people not donating blood and CSF are to be excluded) and/or might decline to take part. 7. Will the authors have information regarding the headline demographics of the memory clinic population as a whole so they know how the Biobank population compares? 8. When the authors note they are the largest memory service in Ireland are they including Northern Ireland in this? If not they should perhaps specify the largest in the Republic of Ireland. 9. Further information regarding the method of CSF collection eg drip method and the make and size of tubes the additional 5-10mls of CSF is collected into would be helpful. 9. A very minor point but how will cognitive symptoms present long-term be recorded, ie when they are not recently different but have been present for a long time. Many people with non-dementia causing diseases will report long-term cognitive symptoms. Thank you
--	--

VERSION 1 – AUTHOR RESPONSE

Reviewer: 1

Dr. Emma Cunningham, Queen's University Belfast

Comments to the Author:

This manuscript helpfully describes in detail the planned set up of a Biobank and will be of interest to clinicians and researchers considering similar endeavours and also for reviewers and readers of subsequent outputs from this Biobank.

- Many thanks for your comments and both careful and insightful review of our protocol which we have now amended in line with your suggestions as outlined below. Changes are made in the main text in red font.

I have a few suggestions:

1. Information regarding the recruitment timeline could be made more clear. The text suggests people are approached at the time of LP/phlebotomy but it also says phlebotomy can be undertaken for the sole purpose of Biobanking. It would seem likely that phlebotomy occurs on the same day as the clinical assessment that all the clinical data is collected at, are people approached then? Is inclusion based on agreeing to donate a blood or CSF sample - I assume this is the case but this should be explicitly stated. Who are the study investigators/sub-investigators, are they members of the direct clinical care team?

- Thanks, we agree this could be made clearer. Individuals are invited to participate at time of clinical assessment in our MASS. Following this, individuals undergoing lumbar puncture as part of diagnostic workup are invited to participate with lumbar puncture and blood sampling performed on the same day alongside their clinical appointment. Individuals not undergoing lumbar puncture are invited to give a separate blood sample (incorporated alongside routine phlebotomy where possible). We have now updated the text to make this clear

2. It would be nice if the consent form and if possible the Participant Information Leaflet could be included as supplementary material.

- Thanks, we agree and this is now included as approved by our ethics committee

3. The text below Table 1 includes clinical information not included in Table 1. Conversely, page 14 line 46 to page 15 line 13 describe assessments that are all included in Table 1 and the text could be omitted if the references were included in the Table.

- Thanks, we have now changed this – it was left over from a previous version. Thanks for spotting this.

4. Would the authors consider including information regarding physical activity, marital status and social support services (this may well be evident from the other information collected but I cannot see it specifically recorded in Table 1)? Driving is also not listed in Table 1 but is mentioned in the text below.

- Thanks, we do record marital status on the case report form which has now been added to Table 1 alongside driving status. Physical activity is not included in the case report form at present, but we will strive to include this in future versions.

5. Will there be the option to retrospectively interrogate participants' medical records to find additional information if this is required subsequent to the baseline and/or 18 month follow up visit? For example if routine blood biochemistry results are recorded as abnormal yes/no would it be possible to retrieve GFR at the time of blood-based biomarker sampling subsequently? Are there any electronic health records that the Biobank will link to?

- There are no electronic health records in our institution at present, although this is planned in future. The ethics covers all data obtained as part of routine clinical care and this includes blood test results which can be retrieved in terms of specific values depending on the research question as well as a binary normal/abnormal. Both continuous (in standard units recorded in the hospital laboratory) and the binary outcome (normal/abnormal) are now included in Table 1 to add clarity to this point. Many thanks.

6. The authors note they assess 400-500 patients per year and anticipate recruiting 1000 people over 5 years ie 40-50% of those attending memory clinic. Given their wide eligibility criteria can they comment on this anticipated figure. It is a very reasonable figure but would be helpful to understand the reasons the authors think people might not be approached (eg if people not donating blood and CSF are to be excluded) and/or might decline to take part.

- We have estimated this figure based on previous recruitment at our site. We perform around 150 dedicated lumbar punctures per year and assess a further 250 participants a year whom do not undergo lumbar puncture. We estimate that 100 participants of those undergoing lumbar puncture and those not undergoing lumbar puncture would opt to donate samples and this is where we arrived at 200 per year for five years to give 1,000. We have now entered these details in the main text under Study Design.

7. Will the authors have information regarding the headline demographics of the memory clinic population as a whole so they know how the Biobank population compares?

- Unfortunately we do not have this information to hand at present, but will be available annually from the biobank as it progresses and data is entered on the RedCap system

8. When the authors note they are the largest memory service in Ireland are they including Northern Ireland in this? If not they should perhaps specify the largest in the Republic of Ireland.

- Thanks, now changed to Republic of Ireland throughout.

9. Further information regarding the method of CSF collection eg drip method and the make and size of tubes the additional 5-10mls of CSF is collected into would be helpful.

- Thanks, it is by drip method into 2.5mLs sterile polypropylene tubes which are entirely filled. These are made by Starstedt, Cat no 63.614.625 now included in text

9. A very minor point but how will cognitive symptoms present long-term be recorded, ie when they are not recently different but have been present for a long time. Many people with non-dementia causing diseases will report long-term cognitive symptoms.

- This is an important point, we record the main symptoms that are present – i.e the reason for presentation and how long the new symptoms have been present for. Our comprehensive checklist of symptoms aims to capture this in detail, which is paired with neuropsychological assessment.

VERSION 2 – REVIEW

REVIEWER	Emma Cunningham Queen's University Belfast, Centre for Public Health
REVIEW RETURNED	13-Nov-2023
GENERAL COMMENTS	There are some typographical errors in a new sentence in the discussion section "On of the skey strenghs".